# Causal Discovery with Heterogeneous Observational Data

**Fangting Zhou**[1,2]         **Kejun He**[2]         **Yang Ni**[1]

[1]Department of Statistics, Texas A&M University, College Station, Texas, USA
[2]Institute of Statistics and Big Data, Renmin University of China, Beijing, China

## Abstract

We consider the problem of causal discovery (structure learning) from heterogeneous observational data. Most existing methods assume homogeneous sampling scheme and causal mechanism, which may lead to misleading conclusions when violated. We propose a novel approach that exploits data heterogeneity to infer possibly cyclic causal structures from causally insufficient systems. The core idea is to model the direct causal effects as functions of exogenous covariates that help explain sampling and causal heterogeneity. We investigate the structure identifiability properties of the proposed model. Structure learning is carried out in a fully Bayesian fashion, which provides natural uncertainty quantification. We demonstrate its utility through extensive simulations and two real-world applications.

## 1 INTRODUCTION

Causal discovery is a central task in various fields including social science, artificial intelligence, and systems biology. While randomized controlled trials are the gold standard to establish causality, they can be too costly, unethical, or impossible to carry out. For example, recovering gene regulatory networks through gene knockout would be too expensive to scale whereas observational genomic data are considerably easier to obtain with next-generation sequencing technologies and have become widely available. Many causal discovery methods, therefore, attempt to discover causality from purely observational data.

**Related work**  One prominent approach in presenting and learning causality is to use the structural equation model (SEM) and the associated causal graph [Pearl, 1998]. The recursive linear Gaussian SEM is among the most popular

ones although the associated causal directed acyclic graph is only identifiable up to Markov equivalence classes [Verma and Pearl, 1990]. In order to uniquely identify causal structures with observational data, additional distributional assumptions have been made in prior works including the linear non-Gaussian model [Shimizu et al., 2006], the nonlinear additive noise model [Hoyer et al., 2008a], and the linear Gaussian model with equal error variances [Peters and Bühlmann, 2014]. A common thread of these methods is that they assume that the causal graph is acyclic and there are no unmeasured confounders (also known as causal sufficiency). However, directed cycles and confounders are very common in practice. For example, feedback loops (directed cycles) are common regulatory motifs in biological signaling systems [Brandman and Meyer, 2008]. As for confounders, gene regulation is known to be affected by many factors such as epigenetic modification [Portela and Esteller, 2010], which may not be measured together with gene expressions.

To allow for cycles, non-recursive SEMs have been developed and proven to be identifiable for linear non-Gaussian models [Lacerda et al., 2008] and nonlinear additive noise models [Mooij et al., 2011]. In the presence of unmeasured confounders, linear non-Gaussian SEMs have received lots of attention: various models have been proposed and shown to be structurally identifiable under the respective confounding assumptions [Hoyer et al., 2008b, Chen and Chan, 2012, Shimizu and Bollen, 2014, Salehkaleybar et al., 2020]. Nevertheless, none of the aforementioned methods explicitly deal with and provide identifiability guarantees for graphs with both cycles and confounders. Although Hyttinen et al. [2012] and Forré and Mooij [2018] provided learning algorithms for general SEMs (allowing cycles, confounders, and nonlinearity), the graph structure can only be fully recovered with interventional data, which is quite different from the observational setting considered in this paper.

Furthermore, all the aforementioned methods assume independent and identically distributed (iid) observations, which may be violated in many applications. For example, can-

*Accepted for the 38th Conference on Uncertainty in Artificial Intelligence* (UAI 2022).

cer is known to be a genetically heterogeneous disease and, therefore, cancer genomic data exhibit great heterogeneity. Methods that ignore such heterogeneity can perform poorly as we will see in our experiments. Recently, Ni et al. [2019], Huang et al. [2020], Saeed et al. [2020] explicitly addressed the heterogeneity issue by incorporating covariates or using a latent mixture model but their models are acyclic and do not account for unmeasured confounders. The method proposed by Peters et al. [2016] is able to identify the causal relationships in heterogeneous data if the causal effects are invariant across environments. By contrast, we assume the causal mechanism varies with the environment. Our motivating application in cancer genomics is one example where causal variance is more likely to hold than invariance because gene regulation may change as cancer progresses [Moustakas and Heldin, 2007, Huang et al., 2009]. Other examples include finance data (e.g., stock prices) where causal relationships can change over time, and fMRI data where brain connectivity networks can change from subject to subject. Mooij et al. [2020] proposed a flexible joint causal inference (JCI) framework, which allows the causal mechanism to vary. Faria et al. [2022] dealt with discrete groups of interventional samples with known interventional targets but they did not provide causal identification guarantee. Creager et al. [2021] focused on invariant learning and environment inference in tasks like domain generalization, which is related to but different from causal discovery.

In this paper, we propose a novel method for **C**ausal discovery with **H**eterogeneous **O**bservational **D**ata (CHOD). Importantly, we do not restrict our model to be acyclic and do not assume causal sufficiency. By exploiting the data heterogeneity via exogenous covariates, we provide sufficient conditions under which CHOD is structurally identifiable in (i) causally insufficient bivariate cyclic graphs, (ii) causally insufficient multivariate acyclic graphs, and (iii) causally sufficient multivariate cyclic graphs. Our method is among the first model-based causal discovery methods to identify causal graphs with both cycles and confounders in purely observational settings without prior domain knowledge. Extensive simulation experiments and two real-world applications support the utility of our method and demonstrate its superiority in handling heterogeneous data through comparison with state-of-the-art alternatives.

## 2 PRELIMINARIES OF CAUSAL DISCOVERY

Let $\boldsymbol{X} = (X_1, \ldots, X_p)^T$ be a $p$-dimensional random vector. We represent the causal structure as well as the joint observational distribution of $\boldsymbol{X}$ by a linear SEM, $\boldsymbol{X} = \boldsymbol{B}\boldsymbol{X} + \boldsymbol{E}$, with direct causal effects $\boldsymbol{B} = [b_{j\ell}] \in \mathbb{R}^{p \times p}$ and random noises $\boldsymbol{E} = [e_j] \in \mathbb{R}^p$. If $b_{j\ell} \neq 0$, then $X_\ell$ is a direct cause of $X_j$. We assume $\boldsymbol{E}$ to be centered Gaussian with covariance $\boldsymbol{S} = [\sigma_{j\ell}]$. When there are no un-

measured confounders (i.e., hidden common causes), the noises are independent of each other and hence $\boldsymbol{S}$ is diagonal. However, the presence of confounders would correlate the noises, making the off-diagonal elements of $\boldsymbol{S}$ non-zero and resulting in a causally insufficient system. To see that, suppose we explicitly model unmeasured Gaussian confounders $\boldsymbol{L}$ via $\boldsymbol{X} = \boldsymbol{B}\boldsymbol{X} + \boldsymbol{\Gamma}\boldsymbol{L} + \boldsymbol{E}'$. Then marginalizing out $\boldsymbol{L}$ leads to $\boldsymbol{X} = \boldsymbol{B}\boldsymbol{X} + \boldsymbol{E}$ where $\boldsymbol{S} = \text{Cov}(\boldsymbol{E}) = \boldsymbol{\Gamma}\text{Cov}(\boldsymbol{L})\boldsymbol{\Gamma}^T + \text{Cov}(\boldsymbol{E}')$. Therefore, the non-zero off-diagonal entries of $\boldsymbol{S}$ implicitly account for unmeasured confounders.

We use a *mixed graph* $\mathcal{G}_M = (V, E_B, E_D)$ to represent the causal relationships embedded in the SEM, where $V = \{1, \ldots, p\}$ is the set of *nodes* representing $\boldsymbol{X}$, $E_B$ is the set of *bidirected edges*, and $E_D$ is the set of *directed edges*; see Figure 1 for a few examples. There is a bidirected edge $\ell \leftrightarrow j$ if $\sigma_{j\ell} \neq 0$, and a directed edge $\ell \to j$ if $b_{j\ell} \neq 0$. In the former case, $X_j$ and $X_\ell$ are confounded by at least one hidden common cause, while in the latter case, $X_\ell$ is a direct cause of $X_j$. The graph is acyclic if there does not exist a directed path $k_1 \to k_2 \to \ldots \to k_\ell \to k_1$ that returns a node to itself, otherwise it is called cyclic. Our goal is to identify the edge-induced subgraph $\mathcal{G} = (V, E_D)$ with direct causal relationships $E_D$ among the observed variables $\boldsymbol{X}$.

## 3 METHOD

### 3.1 PROPOSED MODEL

Our key idea to discover causality is to take advantage of the data heterogeneity, which we assume can be explained by some exogenous covariates $\boldsymbol{Z} \in \mathbb{R}^q$. The exogenous covariates may be observed (e.g., biomarkers in cancer genomic data) or latent. In the latter case, one can impute the latent covariates by various embedding methods such as t-SNE [van der Maaten and Hinton, 2008] and UMAP [McInnes et al., 2018]. Alternatively, latent covariates can be learned simultaneously with our model. For ease of exposition, we first focus our discussion on the case where the exogenous covariate is given (either observed or imputed) and univariate (i.e., $q = 1$), and later briefly discuss the extension to multivariate latent covariates. Specifically, given $Z$, we model $\boldsymbol{X}$ as a varying-coefficient linear Gaussian SEM,

$$\boldsymbol{X} = \boldsymbol{B}(Z)\boldsymbol{X} + \boldsymbol{E}, \quad \boldsymbol{E} \sim N(\boldsymbol{0}, \boldsymbol{S}), \quad (1)$$

where $\boldsymbol{B}(Z) = [b_{j\ell}(Z)] : \mathbb{R} \mapsto \mathbb{R}^{p \times p}$ is a matrix-valued function of $Z$, which characterizes the changes of the direct causal effects with respect to $Z$. Because each observation potentially has a different value of covariate $Z$, the direct causal effects $\boldsymbol{B}(Z)$ are heterogeneous and observation-specific. Note that since $\boldsymbol{S}$ does not depend on $\boldsymbol{Z}$, we implicitly assume that the confounding effects are not heterogeneous. Model (1) implies the conditional distribution of

$X$ given $Z$,

$$\mathbb{P}(X|Z, B, S) = \det(I - B(Z))N((I - B(Z))X|0, S).$$

When $B(Z)$ is constant in $Z$, model (1) is reduced to an ordinary linear Gaussian SEM and hence its underlying causal graph $\mathcal{G}$ is not identifiable. However, as we will show later, the causal graph of model (1) is in general identifiable.

**Relation to existing methods** While most existing causal discovery methods that are applicable to heterogeneous data assume the exogenous covariate to be discrete and finite (i.e., multiple contexts, domains, or experimental conditions), under our framework the exogenous covariate $Z$ can be either continuous or discrete. To match the case of our real-data application (see Section 4.2) and emphasize the advantage of CHOD, we present $Z$ as a continuous covariate in this paper. With continuous $Z$, the proposed method is particularly useful when the data are heterogeneous but there are no clear predefined discrete groups. If the covariate is categorical, the proposed model can be thought of as multi-domain/group-specific graphical models (see, for example, Yajima et al. [2015], Ghassami et al. [2018], Ni et al. [2018]) by viewing the categorical covariate as the domain or group indicator. The proposed model is also reminiscent of causal models with soft interventions by viewing $Z$ as intervention that modifies causal effects; however, the key difference is that our model does not assume the knowledge of the interventional targets (i.e., $Z$ could affect all causal effects) and the interventions are conducted by nature rather than humans. In summary, our model explicitly accounts for the heterogeneity of data generating mechanism via the observation-specific direct causal effects $B(Z)$, which vary smoothly with covariate $Z$. We provide a detailed discussion contrasting the proposed method with two state-of-the-art heterogeneous causal discovery methods from Huang et al. [2020] and Mooij et al. [2020] in Section S1 of the Supplementary Materials.

In the regression context, the varying-coefficient model serves as an important generalization of linear model. As introduced by Hastie and Tibshirani [1993], the class of varying-coefficient models ties together many important structured regression models such as additive models and dynamic linear models into one common framework. Likewise, our proposed model is a natural extension of linear SEMs, which allows varying causal effects. One important ingredient exploited in this paper is that the adoption of varying causal effects helps identify the causal structure. Note that for simplicity, we have assumed linearity and Gaussianity in the current formulation. In addition to efficient computation and causal effect estimation (discussed briefly in Section S2 of the Supplementary Materials), this simple setup allows us to convey the main idea that heterogeneity alone is enough to enable causal identification.

## 3.2 CAUSAL STRUCTURE IDENTIFIABILITY

For model-based causal discovery methods, the non-identifiability issue can be seen from the distributional/observational equivalence point of view. Two CHOD models parameterized by $(B, S)$ and $(B', S')$ are said to be *distributionally/observationally equivalent* if for any values of $(B, S)$ there exist values of $(B', S')$ such that $\mathbb{P}(X|Z, B, S) = \mathbb{P}(X|Z, B', S')$ for all $X$. Clearly, distributionally/observationally equivalent models cannot be distinguished from each other by examining their observational distributions. The causal structure is said to be *identifiable* if there do not exist two distributionally/observationally equivalent causal models such that $\mathcal{G} \neq \mathcal{G}'$.

Throughout, we make the causal Markov assumption [Richardson, 1996], i.e., the probability distribution $\mathbb{P}$ respects the Markov property of the causal graph $\mathcal{G}$. Before stating our main results, we first provide an intuition on how the proposed CHOD is identifiable using a toy example. Consider the bivariate graphs shown in Figure 1. We can distinguish graphs (a)–(b) from graphs (c)–(f) because the marginal variance of $X_2$ is independent of $Z$ in graphs (a)–(b) but depends on $Z$ in graphs (c)–(f) through the causal effect $b_{21}(Z): X_1 \rightarrow X_2$. Likewise, we can separate graphs (c)–(d) from graphs (e)–(f) by examining the marginal variance of $X_1$ which depends on $Z$ through $X_2 \rightarrow X_1$. We may not distinguish (c) from (d) or (e) from (f), but the direct causal relationship between $X_1$ and $X_2$ is determined in either case.

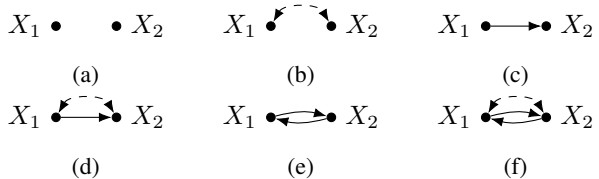

Figure 1: Mixed graphs. Solid arrows are causal effects and dashed bidirected arrows are confounding effects.

We further illustrate the identifiability with simulated data from graphs (b), (d), and (f) in Figure 1. Specifically, the exogenous covariates were generated uniformly. Under each graph, the non-zero elements of $B(Z)$ were assumed to be $0.5\sin(\pi Z)$. We set the noise variances to 1 and the correlation coefficients to 0.5 to have confounding effects. The $n = 1000$ data points as well as the marginal variances estimated by kernel method of the two nodes as functions of $Z$ are depicted in Figure 2 for these 3 cases, from which the causal relationships between $X_1$ and $X_2$ are intuitively identifiable in the presence of both confounders and cycles: in Figure 2(a), both $\text{Var}(X_1)$ and $\text{Var}(X_2)$ are nearly constant in $Z$ indicating no direct causal link; in Figure 2(b), $\text{Var}(X_1)$ is constant but $\text{Var}(X_2)$ is not constant in $Z$ indicating a direct causal link $X_1 \rightarrow X_2$; and in Figure 2(c), neither $\text{Var}(X_1)$ nor $\text{Var}(X_2)$ is constant in $Z$ indicating a

cyclic causal link $X_1 \rightleftarrows X_2$.

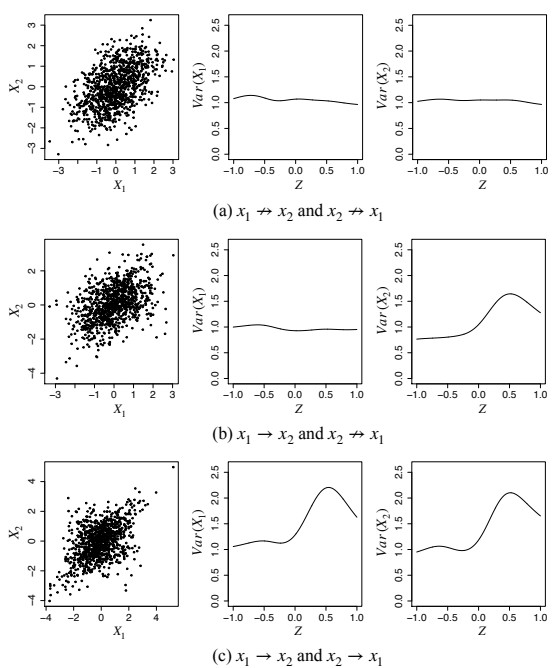

(a) $x_1 \nrightarrow x_2$ and $x_2 \nrightarrow x_1$

(b) $x_1 \rightarrow x_2$ and $x_2 \nrightarrow x_1$

(c) $x_1 \rightarrow x_2$ and $x_2 \rightarrow x_1$

Figure 2: Illustration with a bivariate toy example.

Now we present the identifiability theories. We first show that in the bivariate case the CHOD models are not distributionally equivalent and, therefore, their causal graphs are identifiable.

**Theorem 1** (Causally Insufficient Bivariate Cyclic Graphs). *Consider bivariate CHOD models with direct causal effects $[b_{12}(Z), b_{21}(Z)]$ and $[b'_{12}(Z), b'_{21}(Z)]$, respectively. Assume $b_{j\ell}(Z)$ and $b'_{j\ell}(Z)$ are either zero or non-constant functions for all $j \neq \ell \in \{1, 2\}$. Then if the two CHOD models are distributionally equivalent, we must have $\mathcal{G} = \mathcal{G}'$.*

All proofs are provided in Section S3 of the Supplementary Materials. The assumption that $b_{j\ell}(Z)$ is either zero or a non-constant function is not surprising because if non-zero $b_{j\ell}(Z)$ is constant in $Z$, then the proposed model is reduced to an ordinary linear Gaussian SEM, which is known to be non-identifiable. Loosely speaking, Theorem 1 states that CHOD is identifiable if $Z$ can help explain the heterogeneity of the data generating mechanism.

Next, we provide sufficient conditions for causally insufficient multivariate acyclic systems and causally sufficient multivariate cyclic systems to be identifiable, and leave the theoretical investigation of causally insufficient multivariate cyclic systems as future work. Denote $pa(j) = \{\ell : \ell \rightarrow j \in E_D\}$ as the set of direct causes and $ds(j) = \{\ell : \ell \leftrightarrow \cdots \leftrightarrow j\}$ as the set of nodes connected to $j$ through bidirected arrows.

**Theorem 2** (Causally Insufficient Multivariate Acyclic Graphs). *Consider the CHOD model in (1) restricted to acyclic causal graphs. Assume without loss of generality $(1, \ldots, p)$ is a true causal ordering (i.e., $\ell \nrightarrow j$ if $\ell > j$). If for any node $j$, and any set $S = \{1, \ldots, m\}$ such that $pa(j) \not\subset S$, we have $Var(X_j | \mathbf{X}_S)$ is a non-constant function of the covariate $Z$, then the causal ordering is identifiable. Moreover, if $pa(j) \cap ds(j) = \emptyset, \forall j$, then the causal graph is identifiable.*

The proof of Theorem 2 first identifies an ordering by recursively finding root variables in acyclic graphs and then identifies the graph structure given the ordering. The assumption on $Var(X_j | \mathbf{X}_S)$ means that the heterogeneous causal effects do not accidentally become constant in any paths, which is similar in spirit to the causal faithfulness assumption. See Section S3.2 of the Supplementary Materials for more discussions.

We have presented our theorems in their strongest forms, i.e., full structure identifiability. If some of the causal effects do not vary with $Z$, then their identification is not always guaranteed (they may still be identifiable in some graphs via v-structure and Meek rules). This is similar to other causal models. For example, in additive noise models, all causal effects have to be nonlinear for full identification. Those linear causal effects have to rely on v-structure and Meek rules to achieve identification as in our method. In the linear non-Gaussian acyclic model, all but one noises have to be non-Gaussian. Like our model, violation of these assumptions would lead to partial structure identification. We would like to point out though, under our proposed Bayesian learning framework discussed in Section 3.3, we can assess the credibility of inferred edges via posterior inference: edges that have nearly constant causal effects (e.g., if 95% credible bands of $b_{j\ell}(Z)$, which can be computed from Monte Carlo samples, cover constant functions) are deemed less reliable.

Unlike bivariate graphs, the identifiability results of multivariate cyclic graphs for purely observational data are sparse in the literature with few exception [Lacerda et al., 2008], which assumes causal sufficiency and disjoint cycles. In the following theorem, we also make the same assumptions.

**Theorem 3** (Causally Sufficient Multivariate Cyclic Graphs). *Consider the CHOD model (1). Assume there are no unmeasured confounders and all cycles are disjoint. The causal graph is generally identifiable[1].*

Theorems 1–3 assume $Z$ to be univariate and known (observed or imputed). When $\mathbf{Z}$ is multivariate and unknown, it can be inferred jointly with the causal graph. We provide its identifiability result below and briefly discuss its implementation in Section S4.1 of the Supplementary Materials.

---

[1] That is, it is identifiable unless a peculiar condition holds. We discuss that condition $(\star)$ in Section S3.3 of the Supplementary Materials.

**Proposition 1** (Multivariate Latent Exogenous Covariates). *Assume the vector $\boldsymbol{m}(\boldsymbol{Z})$ that stacks the non-zero elements of $\boldsymbol{B}(\boldsymbol{Z})$ is continuous and injective, and $(\boldsymbol{m}, \boldsymbol{S}) \mapsto \mathbb{P}(\boldsymbol{X}|\boldsymbol{m}, \boldsymbol{S})$ is continuous and injective in $\boldsymbol{m}$ given $\mathcal{G}$. Then the latent exogenous covariates are identifiable up to a monotone transformation.*

Proposition 1 shows that the *relative* position of the latent covariates can be identified, which is useful in sorting observations (see many prominent examples of trajectory inference in single-cell genomic studies [Saelens et al., 2019]). It can be also viewed as an embedding and dimension reduction tool wrapped in a causal model because the dimension of $\boldsymbol{Z}$ is typically much smaller than $\boldsymbol{X}$. The condition of Proposition 1 that assumes $\boldsymbol{m}(\boldsymbol{Z})$ to be injective as a vector-valued function should not be interpreted as a requirement that each individual function has to be injective. For example, if $b_{12}(Z) = (Z + 1)^2$ and $b_{21}(Z) = Z^2$, neither is injective but the resulting $\boldsymbol{m}(Z) = [b_{12}(Z), b_{21}(Z)]$ is injective. Given the causal structure, the second requirement on the distribution $\mathbb{P}$ is equivalent to the identifiability of causal effects or model parameters. The causal effect identification itself is an interesting but challenging task. For linear Gaussian SEMs, it is well-known that causal effects are identifiable without confounders. With confounders, Drton et al. [2011] showed that the acyclic mixed graph needs to be a simple graph. As is evident from the proofs of Theorems 2 and 3, the causal effects under the corresponding assumptions are indeed identifiable for our model. In a related work, Salehkaleybar et al. [2020] showed that the causal effects in the presence of latent confounders are identifiable with mild structure assumptions in the non-Gaussian setting. This paper focuses on investigating causal structure identifiability; establishing causal effect identifiability theory for the causally insufficient multivariate cyclic graphs will be an interesting future work.

### 3.3 BAYESIAN STRUCTURE LEARNING

We learn the causal structure through a Bayesian approach by assigning priors on the space of graphs and model parameters. We model the direct causal effects by cubic B-splines with evenly spaced knots $b_{j\ell}(Z) = \sum_{k=1}^{K} \beta_{j\ell k}\phi_k(Z)$, where $\{\phi_k(Z)\}_{k=1}^{K}$ is the set of spline basis. To encourage graph sparsity, a spike-and-slab prior is assigned to the vector $\boldsymbol{\beta}_{j\ell} = (\beta_{j\ell 1}, \ldots, \beta_{j\ell K})^T$,

$$\mathbb{P}(\boldsymbol{\beta}_{j\ell}|r_{j\ell}, \tau) = (1 - r_{j\ell})\delta_{\mathbf{0}}(\boldsymbol{\beta}_{j\ell}) + r_{j\ell}N(\boldsymbol{\beta}_{j\ell}|\mathbf{0}, \tau\boldsymbol{I}),$$

where $\delta_{\mathbf{0}}(\cdot)$ is a point mass at vector zero and $r_{j\ell}$ is a binary edge indicator. By construction, $r_{j\ell} = 0$ if and only if $\boldsymbol{\beta}_{j\ell} = \mathbf{0}$ (equivalently, $\ell \nrightarrow j$ and $b_{j\ell}(Z) \equiv 0$). We assume independent beta-Bernoulli priors with $r_{j\ell} \sim \mathbb{P}(r_{j\ell}|\pi) = $ Bernoulli$(r_{j\ell}|\pi)$ and $\pi \sim \mathbb{P}(\pi) = $ beta$(\pi|a, b)$. We place conjugate inverse-gamma prior on $\tau \sim \mathbb{P}(\tau) = IG(\tau|\alpha, \beta)$ and inverse-Wishart prior on the covariance matrix $\boldsymbol{S} \sim$ $\mathbb{P}(\boldsymbol{S}) = IW(\boldsymbol{S}|\boldsymbol{\Psi}, v)$. If a sparse estimation of confounding effects is desired, selection or shrinkage priors can be assigned to $\boldsymbol{S}$, which we do not pursue in this paper.

Let $\mathcal{D} = \{(\boldsymbol{x}_i, z_i), i \in 1, \ldots, n\}$ be $n$ realizations of $(\boldsymbol{X}, Z)$. Denote $\boldsymbol{\beta} = [\beta_{j\ell k}]$ and $\boldsymbol{r} = [r_{j\ell}]$. The joint posterior distribution is then given by

$$\mathbb{P}(\boldsymbol{\beta}, \boldsymbol{S}, \boldsymbol{r}, \pi, \tau|\mathcal{D}) \propto \mathbb{P}(\boldsymbol{\beta}|\boldsymbol{r}, \tau)\mathbb{P}(\boldsymbol{S})\mathbb{P}(\boldsymbol{r}|\pi)\mathbb{P}(\pi)\mathbb{P}(\tau)$$
$$\times \prod_{i=1}^{n} \mathbb{P}(\boldsymbol{x}_i|z_i, \boldsymbol{B}(z_i), \boldsymbol{S}),$$

where $\mathbb{P}(\boldsymbol{\beta}|\boldsymbol{r}, \tau) = \prod_{j,\ell} \mathbb{P}(\boldsymbol{\beta}_{j\ell}|r_{j\ell}, \tau)$ and $\mathbb{P}(\boldsymbol{r}|\pi) = \prod_{j,\ell} \mathbb{P}(r_{j\ell}|\pi)$. The posterior distribution is not analytically available; we use Markov chain Monte Carlo (MCMC) to approximate it with the sampling steps detailed in Section S4 of Supplementary Materials.

The per-iteration computational complexity of sampling is $O(np^5)^2$. This is a general learning algorithm that includes possible cycles and confounders. It can be simplified if there are no cycles and/or confounders. For example, for acyclic graphs, the spline coefficients $\boldsymbol{\beta}$ can be integrated out to improve MCMC mixing. Upon the completion of MCMC, the causal structure can be summarized by thresholding the estimated posterior probability of inclusion $\mathbb{P}(r_{j\ell} = 1|\mathcal{D}) \approx 1/M \sum_{m=1}^{M} I(r_{j\ell}^{(m)} = 1)$ at 0.5, where the superscript $(m)$ indexes the Monte Carlo samples. Alternatively, we can choose a different threshold to control the Bayesian false discovery rate as in Müller et al. [2006].

## 4 EXPERIMENTS

We use extensive simulations as well as a real cancer genomic dataset with known cyclic causal graphs to evaluate the proposed method, CHOD. Additionally, we also applied CHOD to a Hong Kong stock market dataset analyzed in a recent heterogeneous causal discovery work [Huang et al., 2020]. Throughout, we set the hyperparameters as non-informative ones with $\boldsymbol{\Psi} = \boldsymbol{I}$, $v = p$, $a = b = 0.5$, $\alpha = \beta = 0.01$, and $K = 10$, which performed well in all experiments considered. We ran MCMC for 2000 iterations, discarded the first 1000 iterations as burn-in, and retained every 5th iteration after burn-in as posterior samples. We evaluated the graph structure recovery accuracy by calculating true positive rate (TPR), false discovery rate (FDR), and Matthew's correlation coefficient (MCC) based on 50 repetitions in simulations. TPR (higher is better) measures the sensitivity/power of the method, i.e., how many true edges can a method detect, and FDR (lower is better) measures how many detected edges are false discoveries. A good method should have high TPR and low FDR. MCC (higher is better) is a unified measure that accounts for both TPR and FDR. It takes value in $[-1, 1]$ with 1 indicating perfect graph recovery.

---

[2] Sampling $\boldsymbol{r}$ and $\boldsymbol{\beta}$ requires $O(p^2)$ numbers of likelihood evaluation and each likelihood evaluation is $O(np^3)$.

## 4.1 SIMULATIONS

Since most existing causal discovery methods assume acyclic graphs and/or causal sufficiency with some exceptions that allow either cycles or confounders but usually not both, in order to maximize the fairness of comparison, we conducted simulations under three scenarios: when the simulation truths are cyclic graphs with confounders, acyclic graphs with confounders, and cyclic graphs without confounders, respectively. The first scenario is the most general one, which has been the focus of this paper, while the second and third scenarios are designed for fairness and hence are briefly discussed in the main text with details provided in the Section S5.1 of the Supplementary Materials. Note that our general algorithm accommodates all those three settings. We first considered data generated from our proposed model and then considered various model misspecifications in terms of non-Gaussian errors, different confounding effects, varying degrees of heterogeneity, and unobserved covariates.

**Data generating mechanism** We considered sample size $n \in \{125, 250, 500, 1000\}$ and the number of nodes $p \in \{10, 25, 50\}$. Exogenous covariates were simulated from the uniform distribution $U(-1, 1)$. True causal graphs were generated as Erdős-Rényi random graph with edge probability $1/p$ (plotted in Figure S2-S4 in Section S5 of the Supplementary Materials). When assumed acyclic in the second scenario, the graph is constrained to have no directed cycles. Given the true structure, non-zero direct causal effects were randomly chosen from $f(Z) = 0.8Z, g(Z) = 0.9\cos(\pi Z)$, or $h(Z) = 0.9\tanh(\pi Z)$. We set the diagonal elements of $\boldsymbol{S}$ to 1. We generated the off-diagonal entries of $\boldsymbol{S}$ randomly from $U(-1, 1)$ in scenarios where there are unmeasured confounders, subject to $\boldsymbol{S}$ being positive-definite. Observations were then generated from model (1).

**Scenario 1: cyclic graphs with confounders** To the best of our knowledge, methods that can deal with both cycles and confounders in purely observational data are uncommon. We compared CHOD with two state-of-the-art acyclic causal discovery methods with confounders: RFCI [Colombo et al., 2012] and RICA [Salehkaleybar et al., 2020], and two state-of-the-art cyclic causal discovery methods without confounders: LiNG [Lacerda et al., 2008] and ANM [Mooij et al., 2011]. RICA and LiNG are based on linear non-Gaussian models, while ANM uses nonlinear additive noise models. RFCI imposes no distributional assumptions and outputs a graph containing both directed and bidirected edges (or edges with indeterminate directions). The results are summarized in Table 1. As expected, CHOD was the only approach that could recover the true graph well under this general heterogeneous simulation setting where both cycles and confounders are present. For example, the MCC for all the competing methods was uniformly low for all $(n, p)$ whereas the MCC of the proposed CHOD was always substantially higher and improved as sample size increased as expected.

**Scenario 2: acyclic graphs with confounders** In addition to RICA and RFCI, in this scenario, we compared CHOD with CAM [Bühlmann et al., 2014], GDS [Peters et al., 2014], and RESIT [Peters et al., 2014] as benchmarks although they are not designed for causal discovery in the presence of confounders. These three methods are based on nonlinear additive noise model. Moreover, we combined several bivariate causal discovery methods with CAM as suggested in Tagasovska et al. [2020] by first using CAM to learn a Markov equivalence class and then using IGCI [Janzing and Schölkopf, 2010], EMD [Chen et al., 2014], and bQCD [Tagasovska et al., 2020] to orient edges. These three bivariate causal discovery methods are based on asymmetry between the cause and the effect in terms of certain complexity metrics. In addition, we also compared with NOTEARS [Zheng et al., 2018] and DAG-GNN [Yu et al., 2019], which utilize continuous optimization for directed acyclic graph learning. Results are summarized in Table S3 in Section S5.1 of the Supplementary Materials. In summary, CHOD outperformed all the competing methods: the MCC of CHOD ranged from 0.6 to 0.9 whereas the competing methods had MCC typically $< 0.4$. Moreover, we conducted additional simulations under the scenario of acyclic graphs without confounders. Still, the performance of these methods did not improve much compared to the scenario with confounders because of the heterogeneity, and the proposed CHOD still significantly outperformed them (results provided in Section S5.1 of the Supplementary Materials). Furthermore, in Section S5.1 of the Supplementary Materials, we considered more comparisons with methods that incorporate the covariate $Z$ as an additional node in the causal graph (similar in spirit to the JCI framework [Mooij et al., 2020]). However, these additional comparisons did not show significantly better graph recovery.

**Scenario 3: cyclic graphs without confounders** We compared CHOD with LiNG and ANM. The results are summarized in Table S4 in Section S5.1 of the Supplementary Materials. As in the first two scenarios, CHOD performed significantly better by exploiting the data heterogeneity.

**Misspecification 1: nonlinear confounding, non-Gaussianity, and varying degrees of heterogeneity** From previous experiments, the proposed CHOD consistently outperformed non-Gaussian SEMs because data were heterogeneous and the errors were Gaussian, both conditions favoring CHOD. For fairer comparison and better illustration, we conducted further simulations under an alternative data generating mechanism. Specifically, we mimicked the simulation setting in Salehkaleybar et al. [2020] by generating $n = 250$ observations from the SEM (1) with uniform noises $e \sim U(-1, 1)$ under a three-node (Figure S5(b))

Table 1: Simulation Scenario 1. Average operating characteristics over 50 repetitions. The standard deviation for each statistic is given within parentheses. The best performance is shown in boldface.

| $n = 125$ | $p = 10$ | | | $p = 25$ | | | $p = 50$ | | |
|---|---|---|---|---|---|---|---|---|---|
| | TPR | FDR | MCC | TPR | FDR | MCC | TPR | FDR | MCC |
| CHOD | 0.662 (0.065) | **0.253 (0.092)** | **0.644 (0.077)** | 0.662 (0.062) | **0.385 (0.080)** | **0.590 (0.094)** | 0.608 (0.082) | **0.385 (0.077)** | **0.590 (0.061)** |
| LiNG | **0.913 (0.088)** | 0.863 (0.012) | 0.104 (0.073) | **0.860 (0.057)** | 0.953 (0.003) | 0.023 (0.034) | **0.802 (0.059)** | 0.979 (0.002) | 0.007 (0.021) |
| ANM | 0.093 (0.069) | 0.879 (0.091) | 0.004 (0.083) | 0.002 (0.006) | 0.988 (0.035) | 0.009 (0.015) | 0.002 (0.006) | 0.972 (0.043) | 0.004 (0.022) |
| RFCI | 0.174 (0.069) | 0.742 (0.082) | 0.113 (0.071) | 0.227 (0.035) | 0.715 (0.046) | 0.200 (0.041) | 0.046 (0.025) | 0.943 (0.026) | 0.030 (0.025) |
| RICA | 0.470 (0.130) | 0.895 (0.029) | 0.051 (0.009) | 0.566 (0.104) | 0.947 (0.009) | 0.037 (0.045) | 0.485 (0.044) | 0.978 (0.002) | 0.006 (0.013) |

| $n = 250$ | $p = 10$ | | | $p = 25$ | | | $p = 50$ | | |
|---|---|---|---|---|---|---|---|---|---|
| | TPR | FDR | MCC | TPR | FDR | MCC | TPR | FDR | MCC |
| CHOD | 0.804 (0.081) | **0.162 (0.088)** | **0.768 (0.075)** | 0.698 (0.065) | **0.286 (0.090)** | **0.662 (0.056)** | 0.680 (0.083) | **0.340 (0.081)** | **0.644 (0.086)** |
| LiNG | **0.880 (0.084)** | 0.870 (0.010) | 0.064 (0.068) | **0.834 (0.099)** | 0.955 (0.005) | 0.007 (0.051) | **0.798 (0.047)** | 0.979 (0.001) | 0.004 (0.017) |
| ANM | 0.077 (0.056) | 0.910 (0.057) | 0.024 (0.058) | 0.009 (0.013) | 0.937 (0.048) | 0.010 (0.033) | 0.003 (0.029) | 0.964 (0.012) | 0.004 (0.002) |
| RFCI | 0.201 (0.050) | 0.739 (0.045) | 0.123 (0.047) | 0.280 (0.033) | 0.744 (0.027) | 0.203 (0.030) | 0.009 (0.030) | 0.907 (0.027) | 0.069 (0.028) |
| RICA | 0.463 (0.127) | 0.890 (0.033) | 0.033 (0.099) | 0.481 (0.120) | 0.955 (0.011) | 0.001 (0.051) | 0.472 (0.065) | 0.978 (0.003) | 0.008 (0.019) |

| $n = 500$ | $p = 10$ | | | $p = 25$ | | | $p = 50$ | | |
|---|---|---|---|---|---|---|---|---|---|
| | TPR | FDR | MCC | TPR | FDR | MCC | TPR | FDR | MCC |
| CHOD | 0.849 (0.073) | **0.152 (0.073)** | **0.813 (0.073)** | 0.813 (0.063) | **0.268 (0.090)** | **0.768 (0.094)** | 0.804 (0.075) | **0.259 (0.057)** | **0.768 (0.061)** |
| LiNG | **0.917 (0.099)** | 0.867 (0.013) | 0.093 (0.083) | **0.875 (0.073)** | 0.952 (0.004) | 0.034 (0.039) | 0.788 (0.063) | 0.979 (0.002) | 0.002 (0.022) |
| ANM | 0.100 (0.066) | 0.898 (0.071) | 0.026 (0.074) | 0.009 (0.013) | 0.917 (0.058) | 0.009 (0.041) | 0.004 (0.008) | 0.961 (0.015) | 0.003 (0.018) |
| RFCI | 0.274 (0.045) | 0.734 (0.050) | 0.147 (0.052) | 0.320 (0.043) | 0.772 (0.030) | 0.197 (0.039) | 0.124 (0.027) | 0.901 (0.018) | 0.086 (0.022) |
| RICA | 0.413 (0.126) | 0.903 (0.033) | 0.078 (0.104) | 0.491 (0.094) | 0.954 (0.009) | 0.006 (0.040) | 0.489 (0.082) | 0.977 (0.004) | 0.015 (0.025) |

| $n = 1000$ | $p = 10$ | | | $p = 25$ | | | $p = 50$ | | |
|---|---|---|---|---|---|---|---|---|---|
| | TPR | FDR | MCC | TPR | FDR | MCC | TPR | FDR | MCC |
| CHOD | **0.912 (0.081)** | **0.142 (0.069)** | **0.840 (0.083)** | **0.894 (0.072)** | **0.245 (0.064)** | **0.813 (0.062)** | **0.849 (0.093)** | **0.251 (0.062)** | **0.786 (0.079)** |
| LiNG | 0.897 (0.108) | 0.866 (0.014) | 0.088 (0.089) | 0.836 (0.058) | 0.954 (0.003) | 0.013 (0.030) | 0.814 (0.058) | 0.978 (0.001) | 0.011 (0.020) |
| ANM | 0.127 (0.059) | 0.908 (0.042) | 0.037 (0.055) | 0.010 (0.006) | 0.903 (0.006) | 0.024 (0.006) | 0.003 (0.016) | 0.962 (0.017) | 0.009 (0.017) |
| RFCI | 0.303 (0.043) | 0.754 (0.034) | 0.139 (0.042) | 0.350 (0.028) | 0.792 (0.017) | 0.191 (0.022) | 0.159 (0.022) | 0.897 (0.013) | 0.101 (0.017) |
| RICA | 0.447 (0.142) | 0.893 (0.037) | 0.044 (0.114) | 0.506 (0.073) | 0.951 (0.007) | 0.018 (0.031) | 0.483 (0.078) | 0.977 (0.004) | 0.014 (0.023) |

and a four-node (Figure S5(c)) directed acyclic graph, both having one unmeasured confounder (red, discarded at the model fitting stage). The non-zero direct causal effects were assumed to be quadratic with varying degrees of curvature (Figure S5(a)). As the degree of curvature approached zero, the data became more homogeneous. The coefficient functions were scaled to have the same average effects, that is we kept $\int |b(Z)|dZ$ constant.

We compared CHOD with RICA. Their receiver operating characteristic (ROC) curves under each true graph are shown in Figure S5(d) and S5(e) in the Supplementary Materials, respectively. In both cases, CHOD's performance deteriorated as the degree of heterogeneity *decreased*; by contrast, RICA's performance deteriorated as the degree of heterogeneity *increased*. Not surprisingly, when the data were completely homogeneous, CHOD was no better than a random guess in the three-node graph. However, somewhat surprisingly, in the four-node graph, CHOD still performed reasonably well even when the data were homogeneous. For instance, the worst area under the ROC curve were 0.865 and 0.789 for CHOD and RICA, respectively. Also note that, given sufficient degrees of heterogeneity, CHOD performed reasonably well despite the non-Gaussianity and nonlinear confounding. For further comparison, we reran the analyses on data generated with Gaussian noises (keeping everything

else the same). The results are shown in S5(g) and S5(e) in the Supplementary Materials. RICA were close to random guesses in both graphs whereas CHOD thrived on heterogeneity, especially in the four-node graph.

**Misspecification 2: partially homogeneous data and unknown covariates** We considered another type of model misspecification where data were partially homogeneous (i.e., observations were clustered and iid within each cluster) and the unknown covariate was estimated by UMAP. CHOD significantly outperformed the competing methods (see Section S5.2 of the Supplementary Materials).

## 4.2 APPLICATIONS

**Breast cancer genomic data** We demonstrate the capability of CHOD in identifying gene feedback loops using breast cancer gene expression data downloaded from the Cancer Genome Atlas (https://www.cancer.gov/tcga). Breast cancer is a well-known extremely heterogeneous genetic disease. The dataset contains $n = 1215$ observations with 113 normal and 1102 tumor tissues. We focused on 8 feedback loops involving gene TP53 [Harris and Levine, 2005], plotted in Figure S7 in Section S5.3 of the Supplementary Materials. We compared CHOD with two cyclic causal discovery methods, LiNG and ANM. In addition,

we also considered two versions of JCI Mooij et al. [2020], FCI-JCI and ASD-JCI, which are broadly applicable for causal discovery with heterogeneous data. Gene expressions were log-transformed. For CHOD and JCI, we learned a one-dimensional embedding using UMAP as an input covariate and regressed out the effects of the covariate on the mean gene expression. The results are reported in Table 2. CHOD had uniformly strong performance: it was only outperformed by LiNG in one case in terms of MCC. See Section S5.3 of the Supplementary Materials for a more elaborated description of the comparison.

**Hong Kong stock market dataset** We also applied CHOD to the Hong Kong stock market dataset analyzed by a recent heterogeneous causal discovery paper [Huang et al., 2020], containing 10 major stocks: HSBC Holdings plc (3), Hang Seng Bank Ltd (5), and Bank of China Hong Kong (Holdings) Ltd (10) from Hang Seng Finance Sub-index (HSF); Hong Kong Electric Holdings Limited (4) from Hang Seng Utilities Sub-index (HSU); Cheung Kong Holdings (1), Swire Group (8), and Cathay Pacific Airways Ltd (9) from Hang Seng Commerce & Industry Sub-index (HSC); Wharf (Holdings) Limited (2), Hong Kong Electric Holdings Limited (4), and Sun Hung Kai Properties Limited (7) from Hang Seng Properties Sub-index (HSP). The result is shown in Figure 3, where the causal edges are consistent with some background market knowledge. As indicated by Huang et al. [2020], the within sub-index causal directions $5 \rightarrow 3$, $9 \rightarrow 8$ and $4 \rightarrow 1$ tend to follow the owner-member relationship. In addition, the following findings also match those in Huang et al. [2020]: stocks in HSF are major causes for those in HSC and HSP, and the stocks in HSP and HSU are major causes for those in HSC.

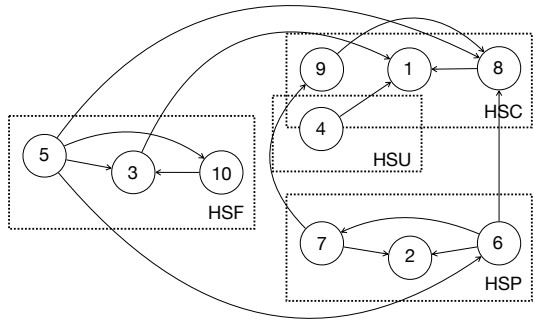

Figure 3: Application to the HK stock market dataset.

## 5 DISCUSSION

We have developed one of the first model-based causal discovery methods for observational data in the presence of both cyclic causality and confounders by exploiting the heterogeneity of causal mechanism. We have established several identifiability theories and carried out extensive ex-

periments to demonstrate the utility of the proposed method against state-of-the-art alternatives.

There are many additional future directions can be taken to extend this work. For example, we have focused on linear Gaussian models, which enable efficient computation and causal effect estimation, and allow us to hopefully have conveyed the main idea that heterogeneity alone is enough to enable causal identification. Since nonlinearity and non-Gaussianity have already been proved useful for causal identification, we might have somewhat masked the contribution of heterogeneity if we had incorporated nonlinearity and/or non-Gaussianity into the proposed model. And as demonstrated in the experiments, the proposed model is relatively robust to non-Gaussian noises. That said, a natural future direction is indeed to extend this paper to nonlinear and non-Gaussian models via e.g., basis expansion and mixture of Gaussian error distributions.

Although we have not proven the most general case with causally insufficient multivariate cyclic graphs, our bivariate result is a strong indicator of identifiability in the general multivariate case because in the multivariate case, the existence of special graph structures (e.g., v-structure) can help causal identification whereas in the bivariate case, we can only rely on cause-effect asymmetry. We plan to prove the general identifiability in the future. Practically, as was done in many previous works like Tagasovska et al. [2020], one may first fix the skeleton or learn some partial structures with common structure learning algorithms, and then orient indeterminate edges by applying the bivariate causal discovery method to identify the full structure.

### Acknowledgements

Ni's research was partially supported by National Science Foundation (DMS-2112943 and DMS-1918851). He's research was partially supported by National Natural Science Foundation of China (No.11801560).

Table 2: Application to breast cancer genomic data. 8 feedback loops involving gene TP53 were considered. The best performance is shown in boldface.

| Method | Network A | | | Network B | | | Network C | | | Network D | | |
|--------|-----|-----|-----|-----|-----|-----|-----|-----|-----|-----|-----|-----|
| | TPR | FDR | MCC | TPR | FDR | MCC | TPR | FDR | MCC | TPR | FDR | MCC |
| CHOD | **0.750** | 0.250 | 0.550 | 0.333 | **0.200** | **0.398** | **0.667** | **0.500** | **0.316** | **0.500** | 0.571 | **0.275** |
| LiNG | **0.750** | **0.000** | **0.791** | 0.583 | 0.563 | 0.198 | 0.333 | 0.667 | 0.000 | **0.500** | 0.667 | 0.164 |
| ANM | 0.500 | 0.333 | 0.316 | **0.667** | 0.556 | 0.238 | 0.333 | **0.500** | 0.189 | 0.167 | **0.500** | 0.180 |
| FCI-JCI | 0.250 | 0.500 | 0.059 | 0.417 | 0.500 | 0.219 | **0.667** | **0.500** | **0.316** | 0.333 | 0.667 | 0.123 |
| ASD-JCI | 0.500 | 0.333 | 0.316 | 0.500 | 0.455 | 0.298 | **0.667** | **0.500** | **0.316** | 0.167 | 0.750 | 0.010 |

| Method | Network E | | | Network F | | | Network G | | | Network H | | |
|--------|-----|-----|-----|-----|-----|-----|-----|-----|-----|-----|-----|-----|
| | TPR | FDR | MCC | TPR | FDR | MCC | TPR | FDR | MCC | TPR | FDR | MCC |
| CHOD | **0.833** | **0.286** | **0.693** | 0.600 | **0.250** | 0.545 | **1.000** | **0.000** | **1.000** | **1.000** | **0.000** | **1.000** |
| LiNG | 0.333 | 0.778 | 0.031 | 0.800 | 0.500 | 0.405 | **1.000** | **0.000** | **1.000** | 0.500 | 0.000 | 0.577 |
| ANM | 0.667 | 0.556 | 0.359 | **1.000** | 0.583 | 0.389 | **1.000** | **0.000** | **1.000** | 0.500 | 0.000 | 0.577 |
| FCI-JCI | 0.500 | 0.700 | 0.114 | 0.400 | 0.667 | 0.035 | 0.000 | - | - | 0.000 | - | - |
| ASD-JCI | 0.500 | 0.625 | 0.217 | 0.600 | 0.571 | 0.221 | 0.000 | - | - | 0.000 | - | - |

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
