# OpenReview forum: "Causal Discovery with Heterogeneous Observational Data"
_auai.org/UAI/2022/Conference — UAI 2022 Poster_

### Official Review · Reviewer_LEtm · 2022-04-11

**Q2(1) Originality/Novelty:** 3
**Q2(2) Significance/Impact:** 2
**Q2(3) Correctness/Technical Quality:** 3
**Q2(6) Clarity Of Writing:** 4
**Q6 Overall Score:** 7
**Q8 Confidence In Your Score:** 3

**Q1 Summary And Contributions:**

The paper presents CHOD, a causal structure learning methodology with heterogeneous observational data. CHOD doesn’t restrict the model to be acyclic and does not assume causal sufficiency. The key assumption is that data heterogeneity depends on specific exogenous covariates Z (known or imputed). Theoretically, the work has shown various identifiability theories (and promises to present general identifiability in the future). The proposed method is implemented on simulated and real datasets.

**Q2 Assessment Of The Paper:**

More detailed information regarding each of these aspects is given below:

**Q2(4) Quality Of Experiments (Optional):**

3: Good: The experimental evaluation is adequate, and the results convincingly support the main claims.

**Q2(5) Reproducibility:**

3: Good: Key resources (e.g., proofs, code, data) are available and key details (e.g., proofs, experimental setup) are sufficiently well-described for competent researchers to confidently reproduce the main results.

**Q3 Main Strengths:**

(1) The background is well-established and the related work section is clear. (2) The figures are explained clearly and aid in understanding the work. (3) The algorithmic methodology is clearly described. (4) The various scenarios to generate different types of causal graphs with observed and unobserved confounders are helpful for other researchers in the future. (5) Validation metrics are well-outlined before experiments.

**Q4 Main Weakness:**

(1) The validation protocol used true positive rate (TPR), false discovery rate (FDR), and Matthew’s correlation coefficient (MCC) to find the efficacy of the proposed method. Since there are no well-established evaluation measures in causal graph generation, some rationale/references would have been helpful.


**Q5 Detailed Comments To The Authors:**

No specific feedback. The paper is well-organized and clearly written.


**Q7 Justification For Your Score:**

The paper in general is well-written and clearly organized. No specific weakness was mentionable, and the strengths, in general, outweigh the weaknesses.

**Q9 Complying With Reviewing Instructions:**

1: Yes.

---

### Official Review · Reviewer_v2z6 · 2022-04-12

**Q2(1) Originality/Novelty:** 3
**Q2(2) Significance/Impact:** 3
**Q2(3) Correctness/Technical Quality:** 4
**Q2(6) Clarity Of Writing:** 3
**Q6 Overall Score:** 9
**Q8 Confidence In Your Score:** 5

**Q1 Summary And Contributions:**

This paper solve the causal discovery problem from heterogeneous observational data, which takes the direct causal effects as functions of exogenous covariates leading to data heterogeneity, and therefore
developed causal discovery methods for observational data in the presence of both cyclic causality and confounders by exploiting the heterogeneity of causal mechanism. They provide solid theory to prove their approach and conduct plenty of experiments to verify their advantage.

**Q2 Assessment Of The Paper:**

More detailed information regarding each of these aspects is given below:

**Q2(4) Quality Of Experiments (Optional):**

4: Excellent: The experimental evaluation is comprehensive and the results are compelling.

**Q2(5) Reproducibility:**

3: Good: Key resources (e.g., proofs, code, data) are available and key details (e.g., proofs, experimental setup) are sufficiently well-described for competent researchers to confidently reproduce the main results.

**Q3 Main Strengths:**

+ The motivation of paper is novel, and problem proposed is important.
+ They propose a sound approach.
+ They provide solid theory to prove their approach.
+ They conduct plenty of experiments to verify their advantage.

**Q4 Main Weakness:**

Some type errors appear.

**Q5 Detailed Comments To The Authors:**

You can improve the presentation to help readers to easy follow this paper.

**Q7 Justification For Your Score:**

The results of paper is significant. It a good paper that benefit to research area.

**Q9 Complying With Reviewing Instructions:**

1: Yes.

---

### Official Review · Reviewer_WjTM · 2022-04-12

**Q2(1) Originality/Novelty:** 4
**Q2(2) Significance/Impact:** 3
**Q2(3) Correctness/Technical Quality:** 3
**Q2(6) Clarity Of Writing:** 3
**Q6 Overall Score:** 7
**Q8 Confidence In Your Score:** 3

**Q1 Summary And Contributions:**

This paper studies the problem of discovering causal graphs from observational data. The paper assumes heterogeneity in the data generation, i.e., the coefficients in the SEM vary with some exogenous covariates, and explores its use in causal discovery. The paper proves the identifiability for various situations and develops the structure learning algorithm through a Bayesian approach. Experiments are conducted using synthetic data and real-world data.

**Q10 Ethical Concerns (Optional):**

No.

**Q2 Assessment Of The Paper:**

More detailed information regarding each of these aspects is given below:

**Q2(4) Quality Of Experiments (Optional):**

3: Good: The experimental evaluation is adequate, and the results convincingly support the main claims.

**Q2(5) Reproducibility:**

3: Good: Key resources (e.g., proofs, code, data) are available and key details (e.g., proofs, experimental setup) are sufficiently well-described for competent researchers to confidently reproduce the main results.

**Q3 Main Strengths:**

The paper explores the heterogeneity in the data generation which has not been studied before. The results are quite impressive. Under the heterogeneity assumption, the causal graphs are identifiable even with cycles and hidden confounders. The experiment results show that the proposed method significantly outperforms the baselines, including LiNG, ANM and gradient-based algorithms.

**Q4 Main Weakness:**

The paper assumes a linear Gaussian SEM.
It is not clear whether the heterogeneity assumption is widely held in practice, as it contradicts another common assumption, i.e., the invariance assumption.


**Q5 Detailed Comments To The Authors:**

Thresholding is an issue that has been studied in some causal discovery algorithms, e.g., NOTEARS. I wonder whether using a different thresholding strategy other than a constant 0.5 would help improve the performance of the proposed method.

The experiment section does not consider an “acyclic graphs without confounders” scenario. As most baselines are developed under this scenario, the readers are curious about how the proposed method performs compared with the baselines in this scenario.

The paper shows that the proposed method performs well in the heterogeneous setting, and other methods perform well in the homogenous setting. This makes me wonder whether the heterogeneity assumption is testable so that an ensemble method can be used to combine different approaches.


**Q7 Justification For Your Score:**

The causal discovery method proposed by this paper based on the heterogeneity assumption is novel. The paper conducts extensive theoretical and empirical analysis of the proposed method. The results show that the proposed method significantly outperforms the baselines.

**Q9 Complying With Reviewing Instructions:**

1: Yes.

---

### Official Review · Reviewer_xLdc · 2022-04-12

**Q2(1) Originality/Novelty:** 2
**Q2(2) Significance/Impact:** 2
**Q2(3) Correctness/Technical Quality:** 2
**Q2(6) Clarity Of Writing:** 3
**Q6 Overall Score:** 4
**Q8 Confidence In Your Score:** 5

**Q1 Summary And Contributions:**

The paper describes CHOD, a causal discovery method from data which have the same causal graphs, but for which the parameters might vary in different environments and we might not know which setting do the samples come from. The methods works for three settings: (1) Bivariate cyclic case (potentially with latent confounders); (2) ADMGs; (3) cyclic DAGs. In the current paper, method assumes linearity and Gaussianity.

**Q2 Assessment Of The Paper:**

More detailed information regarding each of these aspects is given below:

**Q2(4) Quality Of Experiments (Optional):**

3: Good: The experimental evaluation is adequate, and the results convincingly support the main claims.

**Q2(5) Reproducibility:**

3: Good: Key resources (e.g., proofs, code, data) are available and key details (e.g., proofs, experimental setup) are sufficiently well-described for competent researchers to confidently reproduce the main results.

**Q3 Main Strengths:**

The paper focuses on a setting which has a lot of potential interest, i.e. mixtures of heterogeneous distributions for which we might not know for each sample from which regime it was generated from.

The authors did a good job in the empirical evaluation, although some methods in the cyclic+causally insufficient setting were missing (as explained in the weaknesses)



**Q4 Main Weakness:**

Most theoretical contributions (e.g. Thm 1-3) are quite simple, and I think they could be solved by using other methods (e.g. JCI or CD-NOD). In particular one could add a variable Z and learn a graph over the Xis and Z that I think might provide the same results as these theorems (especially if one were to allow cycles and confounders through sigma-separation, as explained below). I'm also not sure why the authors decided to restrict themselves to the linear Gaussian case in the exposition.

The authors state that they couldn't find methods for confounding and cyclic graphs. In the linear case (which is considered here), d-separation still works for cyclic graphs so the authors could have used http://www.its.caltech.edu/~fehardt/papers/HEJ_UAI2014.pdf. For the nonlinear case, one could use sigma-separation: https://arxiv.org/pdf/1807.03024.pdf, which is implemented also in ASD-JCI.

Some related work is missing https://openreview.net/forum?id=hDrn2Dmb7_I

**Q5 Detailed Comments To The Authors:**

I think the paper focuses on an interesting problem, which is causal discovery from heterogenous datasets, when we might not know where a sample comes from. On the other hand, I think the paper overstates its contributions and does not mention explicitly the assumptions one might need (e.g. same graph in all datasets, stable confounding etc). This in particular is an issue since I think most of the theoretical results (Thm1-3) of the paper might be also achieved with existing methods (e.g. CD-NOD for the non-cyclic case or JCI for all cases) by adding the Z variable, which would also not necessarily require restricting assumptions like Gaussianity or linearity. In my opinion, a discussion of the relation with these works is crucial to contextualize the paper in terms of the related research. This would also include recent work on environment inference (e.g. EIIL https://arxiv.org/abs/2010.07249) and mixtures of interventional distributions (https://openreview.net/forum?id=hDrn2Dmb7_I).

As said in the weaknesses, in the comparison the authors state that they couldn't find methods for confounding and cyclic graphs. In the linear case (which is considered here), d-separation still works for cyclic graphs so the authors could have used http://www.its.caltech.edu/~fehardt/papers/HEJ_UAI2014.pdf. For the nonlinear case, one could use sigma-separation: https://arxiv.org/pdf/1807.03024.pdf (which the authors cite already, so I'm a bit confused how they missed it does exactly this), which is implemented also in ASD-JCI

Minor details:
- since the method assumes linearity, I would change the paper title to include linear somewhere
- "causal discovery is a central task... (in) social science, artificial intelligence and systems biology" add citation
- "recovering gene regulatory networks through gene knockout  would be too expensive to scale... " add citation
- "many causal discovery methods, therefore, attempt to discover causality from purely observational data" arguable, since there are some interventional data with gene knockouts (obviously not for the whole genome) and methods that work on mixtures of observational and interventional data, possibly explain a bit more
- "full structure can only be recovered with interventional data" - often, not always as the sentence suggests, plus the key is in how many interventions
- "all the aforementioned methods assume iid observations", but there are many papers on mixtures of observational and interventional data that do not, e.g. [Eaton&Murphy 2007], GIES, COMBINE, [Hyttinen et al 2014] among others that are not cited
- "We use a mixed graph" - probably a directed mixed graph?
- the definition of Eq (1) is not clear at all - can the nonzero values in B(Z) (the graph)) change across datasets? For example can Z=z1 have b_ij =0 and Z=z2 have b_ij != 0?  Is B essentially the fully connected graph or the union of the graphs in all datasets?
- Can there be self-loops? consider the SCM:  X=ZX + epsilon; Y=ZX+ epsilon, would this be allowed? how could one then learn that X causes Y like in Fig 1 (since they both change)?
- So the confounding is assumed to be stable and independent of Z? This is important to mention explicitly
- is the formula for P(X|Z, B, S) correct? Would't one need to have the inverse of the determinant?
- Relation to existing methods - both CD-NOD and JCI can allow for continuous contexts without any modification, so this is not a novelty of this method, and neither assume knowledge of the intervention targets
- the fact that nature conducts interventions and not humans is not really an important distinction, since the math is the same
- why assume linearity and Gaussianity? what would be require to extend this method?
 - "we make causal Markov assumptions" -typo
- Fig 1 would also be identified with ASD-JCI by adding Z, would you agree?
- "We have developed one of the first model-based causal discovery methods for observational data in the presence of both cyclic causality and confounders" - this is just not true, and in particular not true for the linear case

**Q7 Justification For Your Score:**

I think the paper focuses on an interesting problem. On the other hand, I think the writing has to be improved: the paper overstates its contributions, it does not formalize the theory precisely enough to check if things are corect and does not mention explicitly the assumptions one might need (e.g. same graph in all datasets, stable confounding etc), and how it relates to existing work (e.g. CD-NOD, JCI, EIIL, https://openreview.net/forum?id=hDrn2Dmb7_I).

**Q9 Complying With Reviewing Instructions:**

1: Yes.

---

### Decision · Program_Chairs · 2022-05-15

**Decision:**

Accept (Poster)

**Comment:**

Meta Review: This article demonstrates that when a population consists of a mixture of linear Gaussian models in which the underlying causal graph are the same, but the linear coefficients are functions of a latent variable Z, it is possible to distinguish some members  of a Markov equivalence class from each other, thus getting increased identifiability of causal models in the (conditional on Z) linear Gaussian case, even when there may be cycles and latent confounding. Three of the reviewers considered this a solid accept (overall scores of 7,9,7) and novel (scores of 4,3,3) but one reviewer rated it as borderline reject (overall score 4) and of lower novelty (2).

The reviewer with the low score asked for:
A.	some clarification of how this differed from alternative methods which the authors provided in terms of extra identifiability;
B.	clarification of the model (which was clearly expressed in the paper in an equation form, but admittedly could use some explanation of the equation);
C.	experimental comparisons with more algorithms (including ASD-JCI and Faria et al.) which the authors performed, and got better results than either alternative.
In response to the authors, the reviewer replied “Overall, I still feel that although the method has merits, the changes to make the assumptions, theory and contributions clear in a revised version would be quite substantial.” I disagree with that assessment, and feel the authors have answered all of the major objections.

The paper is a solid contribution about a limited (linear Gaussian conditional on Z) but interesting case, and contains novel results that are stronger than constraint-based methods. The simulation and empirical results are good.